# Enhanced SARS-CoV-2-Specific CD4^+^ T Cell Activation and Multifunctionality in Late Convalescent COVID-19 Individuals

**DOI:** 10.3390/v14030511

**Published:** 2022-03-02

**Authors:** Nathella Pavan Kumar, Kadar Moideen, Arul Nancy, Nandhini Selvaraj, Rachel Mariam Renji, Saravanan Munisankar, Jeromie Wesley Vivian Thangaraj, Santhosh Kumar Muthusamy, C. P. Girish Kumar, Tarun Bhatnagar, Manickam Ponnaiah, Sabarinathan Ramasamy, Saravanakumar Velusamy, Manoj Vasant Murhekar, Subash Babu

**Affiliations:** 1Indian Council of Medical Research (ICMR), National Institute for Research in Tuberculosis, Chennai 600031, India; 2International Center for Excellence in Research, Indian Council of Medical Research (ICMR), National Institute for Research in Tuberculosis, Chennai 600031, India; kadarbinabbas@gmail.com (K.M.); arul.p@nirt.res.in (A.N.); nandhu30797@gmail.com (N.S.); rachelmr1610@gmail.com (R.M.R.); saragamma@gmail.com (S.M.); sbabu@nirt.res.in (S.B.); 3Indian Council of Medical Research (ICMR), National Institute of Epidemiology, Chennai 600077, India; stanjeromie@nieicmr.org.in (J.W.V.T.); santhoshkumar@nieicmr.org.in (S.K.M.); girishmicro@gmail.com (C.P.G.K.); drtarunb@gmail.com (T.B.); manickam@nie.gov.in (M.P.); sabari@nieicmr.org.in (S.R.); saravanan.biostat1985@gmail.com (S.V.); mmurhekar@nieicmr.org.in (M.V.M.)

**Keywords:** COVID-19, SARS-CoV-2, CD4^+^ T cells, cytokines

## Abstract

Background: Examination of CD4^+^ T cell responses during the natural course of severe acute respiratory syndrome coronavirus-2 (SARS-CoV-2) infection offers useful information for the improvement of vaccination strategies against this virus and the protective effect of these T cells. Methods: We characterized the SARS-CoV-2-specific CD4^+^ T cell activation marker, multifunctional cytokine and cytotoxic marker expression in recovered coronavirus disease 2019 (COVID-19) individuals. Results: CD4^+^ T-cell responses in late convalescent (>6 months of diagnosis) individuals are characterized by elevated frequencies of activated as well as mono, dual- and multi-functional Th1 and Th17 CD4^+^ T cells in comparison to early convalescent (<1 month of diagnosis) individuals following stimulation with SARS-CoV-2-specific antigens. Similarly, the frequencies of cytotoxic marker expressing CD4^+^ T cells were also enhanced in late convalescent compared to early convalescent individuals. Conclusion: Our findings from a low-to middle-income country suggest protective adaptive immune responses following natural infection of SARS-CoV-2 are elevated even at six months following initial symptoms, indicating the CD4^+^ T cell mediated immune protection lasts for six months or more in natural infection.

## 1. Introduction

The development of severe acute respiratory syndrome coronavirus 2 (SARS-CoV-2) swiftly progressed into a global pandemic. Globally, studies are underway to map the factors of immune protection against SARS-CoV-2 [1]. The clinical spectrum of SARS-CoV-2 infection is extremely flexible, ranging from asymptomatic infection to severe disease [2]. During the course of natural SARS-CoV-2 infection, it is potentially important to monitor the adaptive immune responses, which will provide some beneficial information for the improvement of vaccination strategies against this virus and its emerging variants [3]. As reported previously, similar to other respiratory viral infections, T cells have important role in SARS-CoV-2 infection [4,5]. Nevertheless, it remains imprecise whether T cell responses are completely beneficial or could also be detrimental in COVID-19 infection, and whether both scenarios can occur depending on timing, composition or magnitude of the adaptive immune response.

T cells play a central role in many viral infections, among which CD4^+^ T cells mediate B cell help for antibody production and coordinate the response of other immune cells types, including CD8^+^ T cell lysis of infected cells [6,7,8]. Recent studies have also reported that there was a robust T cell mediated immune response in COVID-19-recovered convalescent individuals [1] and also persistent cellular immunity even after six months of SARS-CoV-2 infection [9]. Knowing the significance of CD4^+^ T cells in anti-viral immunity, examining this adaptive immune cell population will give us more understanding into the type of host responses witnessed in patients with COVID-19 [10]. Recent studies on antigen-specific CD4^+^ T cells phenotyping of SARS-CoV-2-responding cells have systematically captured the coverage of CD4^+^ T cells that respond to SARS-CoV-2 infection [8,10,11,12,13]. Hence, we too wanted to examine a wider panel of T cell mediated immune responses in our study cohort.

The study of the durability of the adaptive immune response in convalescent COVID-19 individuals may accelerate the understanding of how immune protection fosters and continues during the natural course of SARS-CoV-2 infection and, which in turn, provide us some valuable information against this emerging new virus. Published studies have characterized the SARS-CoV-2-specific T cell response in COVID-19 [8,11] and their possible protective role has been gathered from the previous studies of patients who recovered from SARS [14] and MERS [15]. In addition, the persistence of T cell activation and function has been examined mostly in studies in developed countries [3,16,17] and very few studies have addressed this question in low- and middle-income countries [18]. Therefore, from a low-to-middle-income country, we aimed to measure the dynamics and longevity of the SARS-CoV-2-specific immune responses in early and late convalescent COVID-19 individuals.

## 2. Materials and Methods

### 2.1. Study Population

The study was approved by the Ethics Committees of ICMR-NIRT (NIRT-INo:2020047, dated 21 December 2020) and ICMR-NIE (NIE/IHEC/202008-01, dated 19 August 2020). Study participants (*n* = 40) were residents of Chennai and Tiruvallur, *n* = 20 of whom were early convalescent (ECV) COVID-19 or <1 month recovered individuals (15–30 days from RT-PCR confirmation) and *n* = 20 late convalescent (LCV) COVID-19 or >6 months recovered individuals (classified by days from RT-PCR confirmation as more than 180 days from RT-PCR confirmation). Participants over 18 years of age were included in the study between November 2020 and December 2020 after obtaining informed consent. Those who had active COVID-19 infection under home isolation and recovered COVID-19 individuals within 0–15 days of RT-PCR confirmation were excluded from the study. The age group ranged between 18–57 years. COVID-19 was confirmed by RT-PCR in government-approved laboratories. A test result was considered positive if two or more of the SARS genomic targets showed positive results (CT < 45 cycles). The demographic and reported clinical characteristics are the one recorded during their acute disease condition but not during the early or late convalescent period (Table 1).

### 2.2. Cell Preparation

Blood samples were collected from early convalescent or late convalescent individuals. Peripheral blood mononuclear cells (PBMCs) were isolated from heparinized blood by density gradient centrifugation (Ficoll-Paque) and cryopreserved in 90% heat-activated FBS plus 10% DMSO in liquid nitrogen. Thawed PBMCs were washed and resuspended at 2 × 10^6^ cells/mL with RPMI 1640 supplemented with 10% heat-inactivated FBS.

### 2.3. Antigens

SARS-CoV-2 antigens used were PepTivator^®^ SARS-CoV-2 Prot_S, PepTivator^®^ SARS-CoV-2 Prot_S1, PepTivator^®^ SARS-CoV-2 Prot_M and PepTivator^®^ SARS-CoV-2 Prot_N (four from Miltenyi Biotec, Auburn, CA, USA). Pool of lyophilized peptides, consisting mainly of 15-mer sequences with 11 amino acids (aa) overlap, covered the complete sequence of the spike protein (S), membrane glycoprotein (M) and nucleocapsid phosphoprotein (N). SARS-CoV-2 whole cell lysate (from BEI resources, Manassas, VA, USA). Final concentrations were 10 μg/mL for S, S1, M and N and SARS-CoV-2 whole cell lysate. Phorbol myristoyl acetate (PMA) and ionomycin, at concentrations of 12.5 ng/mL and 125 ng/mL, respectively, were used as the positive control stimuli.

### 2.4. Cell Stimulation and Intracellular Staining

In total, 5–6 million PBMC cells were stimulated for 12 h with SARS-CoV-2 peptide pools and whole cell lysate in the presence of αCD28/αCD49d costimulatory antibodies (BD FastImmune; BD Biosciences, Franklin Lakes, NJ, USA). A negative control containing PBMCs and costimulatory antibodies (unstimulated condition) from the same subject and PMA/I stimulation as a positive control was also included for each assay. Following stimulation, cells were washed with PBS and surface stained for 30 min in the dark at 4 °C with viability reagent. The cells were then washed and permeabilized with BD Perm/Wash buffer (BD Biosciences) and stained with intracellular markers for an additional 60 min before washing and acquisition. Antibodies containing surface and intracellular markers are FITC-CD8 (SK1), APC R700-CD3 (UCHT1), BV510-CD4 (SK3), PE-CF594-CD56 (NCAM 16.2), BV421-IL2 (5344.111), APC-TNF-a (6401.1111), BV650-IL17A (N49-653), Per-CP-CD69 (L78), BV421-OX40 (CD252), APC-CD38 (HB7), PerCP-Cy5.5- Perforin (Do-G9), BV421-Granzyme-B (GB11), Fixable viability stain 780 (FVS780) from BD Biosciences, Franklin Lakes, NJ, USA, PerCP-Cy5.5-IFN-γ (4S.B3) from eBioscience-Invitrogen, San Diego, CA, USA and Alex-Fluor 647-Granulysin (DH2), BV650-CD107a (H4A3) from Biolegend, San Diego, CA, USA. Fourteen-color flow cytometry was performed on a FACS Cellesta flow cytometer with FACS Diva software v.7 (Becton Dickinson, Franklin Lakes, NJ, USA). The lymphocyte gating was set by forward and side scatter and at least ~100,000 lymphocytes events were acquired. Data were collected and analyzed using Flow Jo 10.7.1 software (TreeStar Inc., Ashland, CA, USA). Multifunctional T cell cytokines expression was calculated using the Boolean gating method. All data are depicted as frequency (percentage) of T cells expressing cytokine(s) and other immune markers. 

### 2.5. Antibody Measurements

Serological testing for antibodies targeting the viral spike protein (IgG (S)) and IgM was performed using YHLO iFlash 1800 Chemiluminescence Immunoassay Analyzer, Shenzhen, China, using iFlash-SARS-CoV-2 IgG (S) and IgM. The cut-off value for SARS-CoV-2 IgG, according to the manufacturer, was the IgM and IgG concentrations more than or equal to 10.00 AU/mL considered as positive and <10.00 AU/mL considered as non-reactive. Circulating neutralizing antibodies levels were measured using SARS-CoV2 Surrogate Virus Neutralization Test Kit, version 2.0, GenScript, Galaxis West Lobby, Singapore, in accordance with the instructions of the manufacturer. The positive cut off and negative cut off for SARS-CoV-2 neutralizing antibody detection are interpreted based on the inhibition rate. The cut-off value for SARS-CoV2 neutralizing antibody detection, according to the manufacturer, the SARS-CoV2 Surrogate Virus Neutralization more than or equal to 20% was considered as positive and <20% was considered as non-reactive.

### 2.6. Statistical Analysis

Data analyses were performed using GraphPad PRISM (GraphPad Software, Inc., La Jolla, CA, USA). Geometric means (GM) were used for measurements of central tendency. Statistically significant differences between two groups were analyzed using the nonparametric two-sided Mann–Whitney *U* test. Multiple comparisons were corrected using the Holm’s correction. *p* values < 0.05 were considered statistically significant. Cluster analysis was done using the R (A Language and Environment for Statistical Computing) software v4.1.2.

## 3. Results

### 3.1. Clinical Characteristics

We included 40 individuals in this study (20 early convalescent, and 20 late convalescent). The median age was 36 years (range 22 years–57 years). In the early convalescent, IgG and neutralizing antibodies are reactive in *n* = 18 and IgM reactive in *n* = 4. In the late convalescent, IgG *n* = 16 and neutralizing antibodies *n* = 20 are reactive and IgM has no individuals that are reactive. A panel of clinical investigations, such as fever, chills, cough, sore throat, runny nose, taste loss, smell loss, joint pain, hypertension, diabetes and asthma, were documented in both the groups (Table 1). Hematology and biochemical parameters were measured, and no statistically significant differences were seen between both the study groups (data not shown).

### 3.2. Enhanced Frequencies of SARS-CoV-2-Specific Activated CD4+ T Cells in Late Convalescent COVID-19 Individuals

We measured the frequencies of CD4^+^ T cells expressing CD69, CD38 and OX40 at baseline (no stimulation) and following stimulation with either SARS-CoV-2-specific antigens or PMA/I. The gating strategy and a representative flow cytometry pseudocolor plots are shown in Figure 1A. As shown in Figure 1B, at baseline there were significantly increased frequencies of CD4^+^ T cells expressing CD69, CD38 and OX40 in late convalescent compared to early convalescent individuals. In response to SARS-CoV-2 S+S1 (Figure 1C), SARS-CoV-2 M+N (Figure 1D) and SARS-CoV-2 WCL (Figure 1E), we observed significantly increased frequencies of CD4^+^ T cells expressing CD69 and CD38 in late convalescent compared to early convalescent individuals. In contrast, no significant differences were seen in the frequencies of CD4^+^ T cells expressing activation markers upon PMA/I stimulation (Figure 1F). Upon stimulation with all SARS-CoV-2 antigens and P/I resulted in significantly in-creased frequencies of activation markers (Appendix A).

### 3.3. Enhanced Frequencies of SARS-CoV-2-Specific Mono-, Dual- and Multi-Functional CD4^+^ T Cells in Late Convalescent COVID-19 Individuals

We used multiparameter flow cytometry to define the frequencies CD4^+^ T cells expressing IFNγ, IL-2, TNFα and IL-17A at baseline and following stimulation with either SARS-CoV-2 antigens or PMA/I. The gating strategy and a representative flow cytometry pseudocolor plot showing the baseline, SARS-CoV-2 antigens and PMA/I stimulated Th1/Th17 cytokines are shown in Figure 2A. As shown in Figure 2B, late convalescent COVID-19 individuals exhibited significantly elevated frequencies of mono-functional Th1 (TNFα expressing) or dual-functional Th1/Th17 (IFNγ/IL-2 or IFNγ/TNFα or IFNγ/IL-17A or IL-2/TNFα or TNFα/IL-17A co-expressing) or multi-functional (IFNγ/IL-2/TNFα) cells at baseline. Similarly, in response to SARS-CoV-2 S+S1 (Figure 2C), SARS-CoV-2 M+N (Figure 2D) and SARS-CoV-2 WCL (Figure 2E), late convalescent COVID-19 individuals exhibited significantly elevated frequencies of mono- or dual-functional Th1/Th17 cells and in the case of SARS-CoV-2 M+N and SARS-CoV-2 WCL, multi-functional Th1 (IFN-γ/IL-2/TNF-α co-expressing) cells as well. In contrast, late convalescent COVID-19 individuals did not exhibit any significant difference in the frequencies of mono-, dual- or triple Th1/Th17 cells in response to control antigen PMA/I (Figure 1F). Upon stimulation with all SARS-CoV-2 antigens and P/I resulted in significantly in-creased frequencies of Mono-, Dual- and Multi-Functional T cells (Appendix A).

### 3.4. Enhanced Frequencies of SARS-CoV-2-Specific CD4^+^ T Cells Expressing Cytotoxic Markers in Late Convalescent COVID-19 Individuals

We measured the frequencies of CD4^+^ T cells expressing perforin, granzyme B, CD107a and granulysin at baseline (no stimulation) and following stimulation with either SARS-CoV-2-specific antigens or PMA/I. The gating strategy and a representative flow cytometry pseudocolor plots are shown in Figure 3A. As shown in Figure 3B, at baseline there were significantly elevated frequencies of CD4^+^ T cells expressing perforin, CD107a and granulysin in late convalescent compared to early convalescent individuals. In response to SARS-CoV-2 S+S1 (Figure 3C), SARS-CoV-2 M+N (Figure 3D) and SARS-CoV-2 WCL (Figure 3E), we observed significantly increased frequencies of CD4^+^ T cells expressing perforin, CD107a and granulysin in late convalescent compared to early convalescent individuals. However, no significant differences were observed between the two groups following PMA/I stimulation (Figure 3F). Upon stimulation with all SARS-CoV-2 antigens and P/I resulted in significantly in-creased frequencies of cytotoxic markers (Appendix A).

### 3.5. Associations between SARS-CoV-2-Specific Antibodies, Multifunctional T Cell Responses and Other Immune Cell Parameters

We wanted to identify correlations between SARS-CoV-2-specific antibodies and other innate cells in individuals with COVID-19. We used Spearman’s correlation coefficients to determine the correlation effect and data were visualized by heat map color intensity with variables being ordered by hierarchical clustering. A multiparametric matrix correlation plot showed strong significant positive correlations between frequencies of multifunctional cells (IFNγ+IL-2+TNFα), dual functional cells (IFNγ+IL-2/IL-2+TNFα) and SARS-CoV-2-specific antibodies (Figure 4). Some negative correlations were also observed between the frequencies of CD4^+^ T cells expressing TNFα+IL-17A or granzyme B and SARS-CoV-2-specific antibodies (Figure 4). In addition, correlation cluster analysis was performed to see the clustering of variables and the dendrogram was plotted. We have used the complete linkage method for this cluster analysis.

Multiparametric matrix correlation plot of immune markers and SARS-CoV-2-specific antibodies in all individuals of early convalescent (ECV) COVID-19 (*n* = 20) and late convalescent (LCV) COVID-19 (*n* = 20). Spearman’s correlation coefficients are visualized by color intensity. *p* values and Spearman r values are ordered by hierarchical clustering.

## 4. Discussion

A better understanding of natural immunity to SARS-CoV-2 is required for the improved expansion of prevention strategies and treatment options for COVID-19. A recent published report recommend that T cells confer protection, whereby SARS-CoV-2-specific memory T cell responses have been validated in the majority of the COVID-19-recovered individuals even in the absence of measurable circulating antibodies [1]. It is also apparent that the important mechanisms of the adaptive immune system requires that either CD4^+^ T cells or CD8^+^ T cells contribute to the control of SARS-CoV-2 infection, indeed more robust clonal expansion of CD8+ T cells is seen in peripheral blood [8,11,19]. However, the precise correlates of immune protection remain unclear [8,11], but the memory immune cells are crucial in protection against COVID-19. The genome of this SARS-CoV-2 encrypts four main structural proteins, including the spike (S) protein, nucleoprotein (N), membrane (M) protein and envelope (E) protein [20]. Studies have reported that a healthy adaptive immune response with presence of spike neutralizing antibodies (Abs) and circulating follicular helper T cells have been seen in individuals who have recovered from the COVID-19 infection [8,21,22].

T cell activation leads to cell surface marker expression and cell proliferation, and CD69 acts as an early activation marker, which expressed rapidly on the surface of T lymphocytes [23]. Various viral and bacterial infection models exhibited elevated CD69 expression on T cells [24,25]. CD38 is a multifunctional transmembrane protein that is broadly expressed in variety of immune cells including T lymphocytes, monocytes, macrophages, dendritic cells and natural killer (NK) cells [26]. Recent studies have reported that CD38 was found to be potent T cell activation marker for acute COVID-19 infection [1]. In addition to this, studies have also reported that at homeostasis high expression frequencies of immune activation markers such as CD38 and CD69 was significantly elevated in severe acute COVID-19 infection in comparison to acute moderate or healthy control by which it is potentially driven by a highly inflammatory environment [1,27]. However, in our cohort, we observed significantly elevated levels of CD69 and CD38 at baseline and upon SARS-CoV-2 S+S1, M+N and WCL in late convalescent individuals compared to early convalescent individuals, indicating that antigen specific expression of CD38 and CD69 reveals a remarkably persistent reliable and multifaceted hyperactivation in recovered COVID-19 individuals. OX40 is a secondary co-stimulatory molecule, which is expressed after 24 to 72 h after activation on activated CD4^+^ and CD8^+^ T cells [28,29]. However, in our cohort, we observed elevations only in the baseline OX40 expressing CD4^+^ T cells but not upon SARS-CoV-2 antigen stimulation.

Currently there is increasing evidence that cellular immunity plays a main role in resolution of COVID-19, but little is known about the persistence of cellular immunity to SARS-CoV-2 [30,31]. This is considered to be the essential factor when studying an individual’s capability to resist a second exposure to the virus. CD4^+^ T cells can provide protection against SARS-CoV [32,33] and other viral pathogens in animal models [34]. Interferons (IFNs) are renowned cytokines for their antiviral effects, which play a key role in viral proliferation, and IFN-γ is an antimicrobial cytokine, upregulated during COVID-19, both locally in the mucosa and systemically [35,36,37]. IL-2 is majorly produced by CD4+ and CD8+ T cells, as well as some B cells and dendritic cells; its important function is to foster the proliferation of CD4+ and CD8+ T cells [38] Few recent studies have reported that higher levels of IL-2 was seen in asymptomatic and mild COVID-19 groups [5,39]. TNF-α is important cytokine for acute inflammatory reactions by functioning as an amplifier of inflammation [40,41]. Published reports explain there was an altered TNF-α response seen during COVID-19 infection [40] IL-17 play a key role in the pathogenesis of multiple inflammatory and autoimmune disorders [42]. Studies have also shown that increased levels of IL-17A are a silent amplifier of the COVID-19 immune responses [43]. In our study, we evaluated the dynamics and durability of SARS-CoV-2 antigen specific T cell responses in early and late COVID-19 convalescent individuals. We observed that late convalescent COVID-19 infected individuals showed higher proportion of CD4^+^ T cell expressing mono, dual and multi-functional Th1 and Th17 cytokines in response to SARS-CoV-2-specific antigens in comparison to early convalescent COVID-19 infected individuals. Thus far, reported studies focused mainly on the longevity of the specific antiviral Ab response. However, the expansion of multi-functional T cells is important for long-term protection, and the longitudinal dynamics of these multi-functional T cells remain poorly understood. Our findings suggest that the majority of individuals, irrespective of disease severity, can mount specific multi-functional T cell responses, which remain present at more than 6 months post-symptom onset. These findings along with recent studies from COVID-19-recovered individuals show persistent multi-functional SARS-CoV-2 antigen–specific memory, suggest that these cells could contribute to rapid recall responses [16]. Few other studies also reported that convalescent-phase SARS-CoV-2-specific T cells generate broader and persistent immune responses even when neutralizing antibodies declined [1,17,44] and these T cell reposes were maintained at least 10 months after infection [3,17]. In addition to this, our findings also suggest that there was a good immune correlation observed between the immune parameters and SARS-CoV-2-specific antibodies. Therefore, our data propose that a highly functional CD4^+^ T cell response persists in COVID-19 convalescent individuals and possibly contributes to immune-mediated long-term protection.

Our study also sought to explore the distribution and function of CD4^+^ T cells expressing cytotoxic molecules such as perforin, granzyme B, CD107a and granulysin in COVID-19 convalescent individuals. Perforin/granzyme-induced apoptosis is a pathway used by cytotoxic lymphocytes to eliminate infected cells [45]. CD107a (LAMP-1) is a marker for degranulation of activated CD8^+^ T cells and to some extent to CD4^+^ T cells. Studies have reported that CD107a is upregulated on the cell surface upon activation of CD8^+^ T cells [40,46]. Granulysin is a cytolytic and proinflammatory molecule constitutively expressed by NK cells and after activation in both CD8^+^ and CD4^+^ T lymphocytes [47]. It has been reported that during viral infections CD4^+^ T cells expressing perforin can play either a protective and/or pathogenic role [48,49]. Therefore, we measured the expression of cytolytic markers in our cohort. Our findings revealed an elevated frequency of cytolytic markers such as perforin, CD107a and granulysin upon antigen stimulation in late convalescent COVID-19 individuals. Whether these cytolytic molecule containing CD4^+^ T cells contribute to either the disease pathology or protection needs to be explored. One of the major limitation of our study is that individual patients have not been followed up longitudinally for the two time points; we cannot distinguish between the presence and quality of antigen-specific cells and, finally, our study also suffers from being a descriptive study.

Together, our findings provide a functional and phenotypic map of SARS-CoV-2-specific T cell immunity across the COVID-19-recovered individuals, suggesting the presence of elevated frequencies of SARS-CoV-2-specific T cells in the majority of individuals six months after infection. Our findings also suggest that the persistence of activated and multi-functional SARS-CoV-2-specific CD4^+^ T cells is one of the hallmarks of immunity in recovered COVID-19 individuals and could possibly act as a correlate of protective immunity against re-infection. 

## Figures and Tables

**Figure 1 viruses-14-00511-f001:**
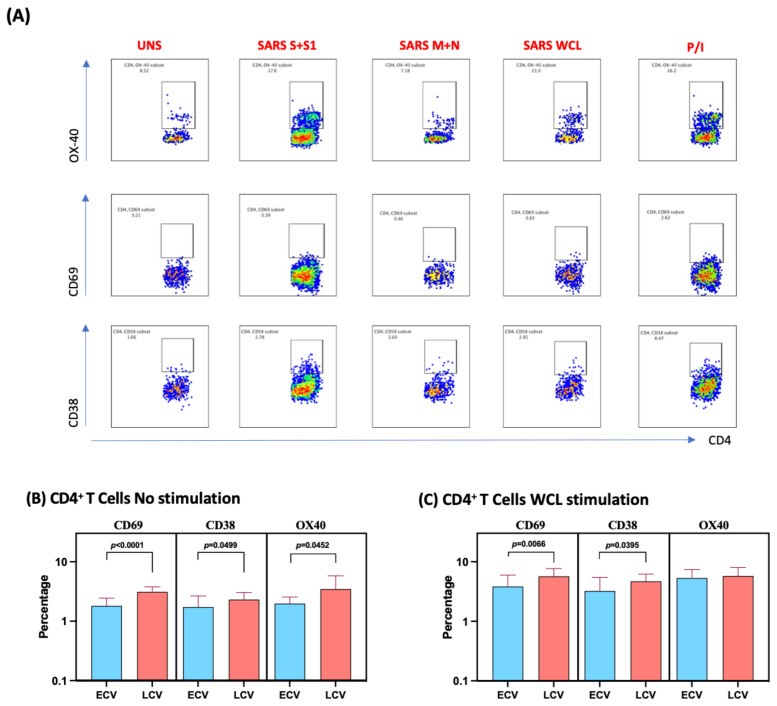
Enhanced frequencies of CD4^+^ T cells expressing activation markers in late convalescent COVID-19 individuals. PBMCs were cultured with media alone or SARS-CoV-2 or control antigens for 12 h and the baseline and antigen-stimulated frequencies of CD69, CD38 and OX-40 were determined in early convalescent (ECV) COVID-19 (*n* = 20) and late convalescent (LCV) COVID-19 individuals (*n* = 20). (**A**) Gating strategy and representative plots for CD4^+^ T cells expressing activation markers. (**B**) The frequencies of CD4^+^ T cells expressing activation markers in early and late convalescent individuals at baseline (**B**) as well as in response to stimulation with (**C**) SARS-CoV-2 WCL, (**D**) SARS-CoV-2 S+S1 peptide pools, (**E**) SARS-CoV-2 M+N peptide pools and (**F**) PMA/Ionomycin were measured by flow cytometry. The bars represent the geometric mean values. *p* values were calculated using the Mann–Whitney test. Any comparison that is not labelled with a *p* value is statistically non-significant.

**Figure 2 viruses-14-00511-f002:**
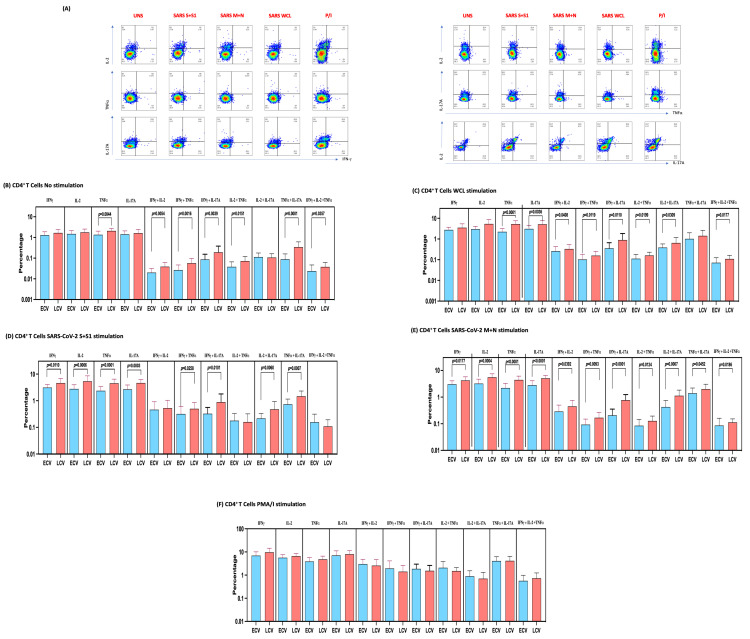
Enhanced frequencies of CD4^+^ T cells expressing multifunctional Th1 and Th17 cytokines in late convalescent COVID-19 individuals. PBMCs were cultured with media alone or SARS-CoV-2 or control antigens for 12 h and the baseline and antigen-stimulated frequencies of multifunctional Th1/Th17 cells were determined in determined in early convalescent (ECV) COVID-19 (*n* = 20) and late convalescent (LCV) COVID-19 individuals (*n* = 20). (**A**) Gating strategy and representative plots for Th1/Th17 CD4^+^ T cell subsets. The frequencies of mono-, dual- and multifunctional CD4^+^ Th1/Th17 cells in early convalescent and late convalescent individuals (**B**) At baseline as well as in response to stimulation with (**C**) SARS-CoV-2 WCL, (**D**) SARS-CoV-2 S+S1 peptide pools, (**E**) SARS-CoV-2 M+N peptide pools and (**F**) PMA/Ionomycin were measured by flow cytometry. The bars represent the geometric mean values. Net frequencies were calculated by subtracting baseline frequencies from the antigen- induced frequencies for each individual. *p* values were calculated using the Mann–Whitney test. Any comparison that is not labelled with a *p* value is statistically non-significant.

**Figure 3 viruses-14-00511-f003:**
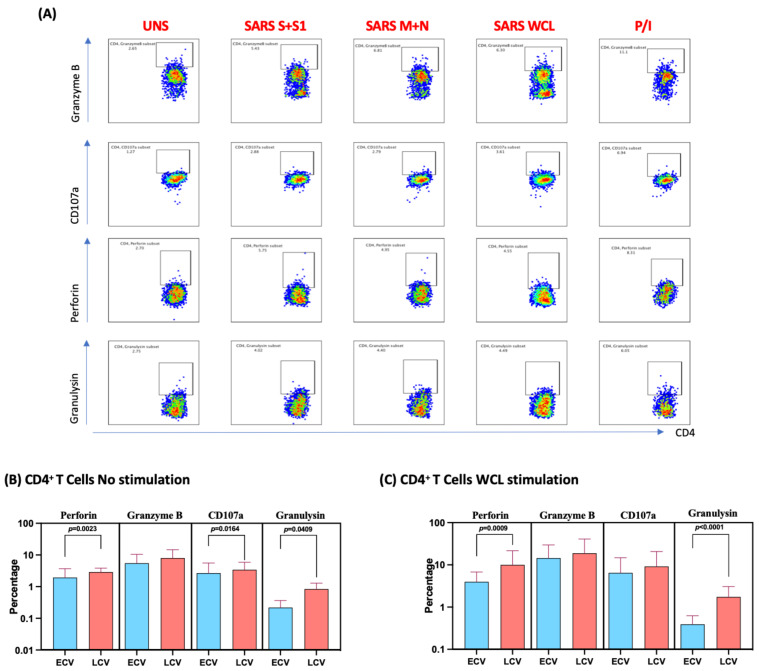
Enhanced frequencies of CD4^+^ T cells expressing cytotoxic markers in late convalescent COVID-19 individuals. PBMCs were cultured with media alone or SARS-CoV-2 or control antigens for 12 h and the baseline and antigen-stimulated frequencies of cytotoxic marker expressing CD4^+^ T cells were determined in early convalescent (ECV) COVID-19 (*n* = 20) and late convalescent (LCV) COVID-19 individuals (*n* = 20). The frequencies of cytotoxic marker expressing CD4^+^ T cells in early convalescent and late convalescent individuals **(A**) Gating strategy and representative plots for CD4^+^ T cells expressing cytotoxic markers. (**B**) at baseline as well as in response to stimulation with (**C**) SARS-CoV-2 WCL, (**D**) SARS-CoV-2 S+S1 peptide pools, (**E**) SARS-CoV-2 M+N peptide pools and (**F**) PMA/Ionomycin were measured by flow cytometry. The bars represent the geometric mean values. Net frequencies were calculated by subtracting baseline frequencies from the antigen-induced frequencies for each individual. *p* values were calculated using the Mann–Whitney test. Any comparison that is not labelled with a *p* value is statistically non-significant.

**Figure 4 viruses-14-00511-f004:**
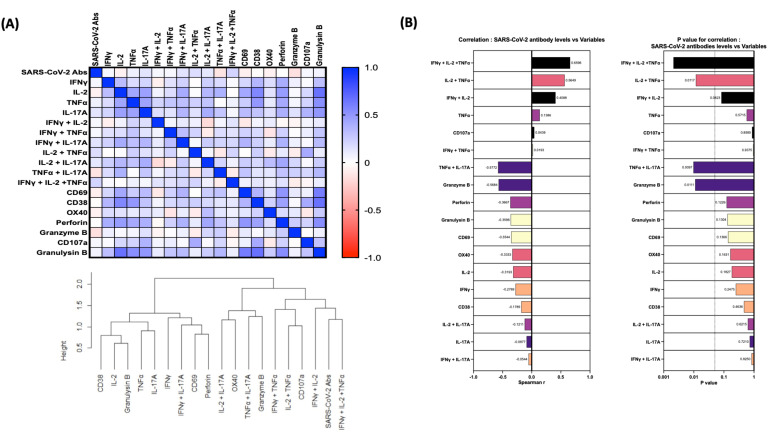
Relationship between Immune markers and SARS-CoV-2 antibodies. (**A**) Multiparametric matrix correlation plot of immune markers and SARS-CoV-2-specific antibodies in all individuals of early convalescent (ECV) COVID-19 (*n* = 20) and late convalescent (LCV) COVID-19 (*n* = 20). (**B**) Spearman’s correlation coefficients are visualized by color intensity. *p* values and Spearman r values are ordered by hierarchical clustering.

**Table 1 viruses-14-00511-t001:** Demographics and clinical symptoms of the study population.

	Early Convalescent	Late Convalescent
Gender (Male/Female)	12/8	13/7
Median age (range)	36 (22–57)	36 (22–57)
Neutralising Antibody (Positive/Negative)	18/2	20/0
IgG (Positive/Negative)	18/2	16/4
IgM (Positive/Negative)	4/16	0/20
Fever	8 (53.3%)	5 (33.3%)
Chills	3 (20%)	2 (13.3%)
Cough	6 (40%)	5 (33.3%)
Sore throat	9 (60%)	3 (20%)
Runny nose	3 (20%)	3 (20%)
Taste loss	8 (53.3%)	4 (26.6%)
Smell loss	6 (40%)	2 (13.3%)
Muscle aches	10 (66.6%)	4 (26.6%)
Joint pain	8 (53.3%)	4 (26.6%)
Hypertension	2 (13.3%)	3 (20%)
Diabetes	1 (6.6%)	3 (20%)
Asthma	1 (6.6%)	0

## Data Availability

All the reported data are available within the manuscript.

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
