# Peer review of "Enhanced SARS-CoV-2-Specific CD4+ T Cell Activation and Multifunctionality in Late Convalescent COVID-19 Individuals"

_viruses, 2022, doi:10.3390/v14030511_

Round 1
Reviewer 1 Report
The work compared the different peptide pools-specific CD4 T cell responses between 1 m patients and 6 m convalescent patients. The authors found that stronger CD4 Th1 and Th17 response and cytolytic biomarkers expression during natural infection. However, the findings was not novel, because the references are not full just like, doi: 10.1093/infdis/jiab543 and doi: 10.1016/j.jaci.2021.09.008, both for 12 m after infection. Second, the methods cannot be performed repeatly, because the describtion is not full to be understood. Major revision is needed to demonstrate the novelty.
- Works should be adjusted to investigate the different peptide pool specific CD4 responses, such as M, N or S or S1 peptide pools specific, respectively, not as just demonstrated M+N pooled specific, or S+S1 pooled, to highlight the role of structural protein specific CD4 druging convalescent.
- Table1 showed something wrong with the cases, just like 5 convalescent patients still have symptoms, which should be exclued from the research. 4 without IgG but nAb present, what kinds of methods have been used for analysis, their detection limitation, and the reason?
- Table 2 should be removed, because no difference between both groups and no medical meaning and all values follow in the normal ranges.
- Figure 1 no BCDEF, all of these should expressed as bar figure and give a statistic analysis between different groups. The same situation for Figure 2 and 3.
- How many PBMC cells used for incubation and FACS?
- Many errors in languages. Just like lines 87-88.
Author Response
The work compared the different peptide pools-specific CD4 T cell responses between 1 m patients and 6 m convalescent patients. The authors found that stronger CD4 Th1 and Th17 response and cytolytic biomarkers expression during natural infection. However, the findings was not novel, because the references are not full just like, doi: 10.1093/infdis/jiab543 and doi: 10.1016/j.jaci.2021.09.008, both for 12 m after infection. Second, the methods cannot be performed repeatly, because the describtion is not full to be understood. Major revision is needed to demonstrate the novelty.
- Works should be adjusted to investigate the different peptide pool specific CD4 responses, such as M, N or S or S1 peptide pools specific, respectively, not as just demonstrated M+N pooled specific, or S+S1 pooled, to highlight the role of structural protein specific CD4 druging convalescent.
Reply: We thank the reviewer for the valuable comment, but unfortunately in our cohort we are been permitted to collect only a minimal volume of blood sample. Hence, we are able to limit the antigenic stimulation by combining the peptide pools.
- Table1 showed something wrong with the cases, just like 5 convalescent patients still have symptoms, which should be exclued from the research. 4 without IgG but nAb present, what kinds of methods have been used for analysis, their detection limitation, and the reason?
Reply: We apologize for the lack of clarity, symptoms that are reported are during the acute COVID-19 infection not during the early or late convalescent period. Lines 95-97
- Table 2 should be removed, because no difference between both groups and no medical meaning and all values follow in the normal ranges.
Reply : As suggested we have now removed the table 2
- Figure 1 no BCDEF, all of these should expressed as bar figure and give a statistic analysis between different groups. The same situation for Figure 2 and 3.
Reply : As suggested all the figures are now changed in to bar graphs. Figures 1-3
- How many PBMC cells used for incubation and FACS?
Reply : 5-6 million cells were used for the antigen stimulated invitro cultures. Line 116
- Many errors in languages. Just like lines 87-88.
Reply : As suggested we have modified the language accordingly. Multiple Lines

Reviewer 2 Report
This manuscript investigates the activation marker, cytokine and effector mediator profile of SARS-CoV-2 reactive T cells early and late after infection. A major aim of the study was to investigate the profile of reactive T cells with different markers in order to enable a systematic overview of the different functionalities. The major observation is that late after infection, there are more functional T cells compared to early after infection. In principle, this observation is surprising, given the fact that you would expect and many other groups have shown that there is peak T cell expansion within the first month after infection, and contraction thereafter. In this regard, it is noteworthy that it is important to differentiate between the mere presence of antigen specific T cells and the detection of T cells that are able to react to antigen stimulation (the latter may comprise a subpopulation of the former, which could explain the findings). This is particularly important for CD4 T cells, for which MHC multimers are less widely used compared to CD8 T cells, which can mark antigen specific T cells irrespective of functionality. An additional important limitation of the study includes the fact that it is not the same individuals who are followed up longitudinally. While the data are in principle of interest to the scientific community, the manuscript in its present needs to be revised considerably before it is suitable for publication.
Specific points:
The entire story would be more convincing if the same “enhancement” of functionality could be shown longitudinally for the same individuals. To interpret the “enhancement” of functionality, it would be important to know the actual time points from which the samples were taken. This should be reported for both the early and late time point.
The introduction is not focused in a way that it introduces specifically the research question that was experimentally tackled by the authors, which is a) wider polyfunctionality of CD4 T cells (more markers) b) short-term vs. long-term phenotype and c) investigation of T cell responses to SARS-2 in a non-high-income country. A first open question that is raised in the introduction is whether T cells are protective or can mediate immunopathology (lines 36-39), but this is not investigated in this study. Then it is mentioned that recent studies have only looked at limited markers (lines 45-48), but out of a gigantic pool of T cell studies on SARS-2 only two studies (ref 8, 10) are mentioned. SARS-2 reactive T cells have been in-depth characterized by scRNA seq (see eg Meckiff et al. Cell 2020, Bacher et al. Immunity 2020 or Fischer et al. Nat Comm 2021) and for a plethora of cytokines eg in Rydyznski Moderbacher et al. Cell 2020, to name a few. See Sette & Crotty et al. Cell 2021 for a review of the T cell response to SARS-2. The authors therefore should tone down their statement that they “looked in to a wider panel of T cell mediated immune responses” (line 48). The authors should also correct the false statement that “very limited studies have characterized the SARS-CoV-2-specific T cell response in COVID-19” (lines 52-53).
Fig. 1A, 2A, 3A. These plots are still hard to read. Percentages should be larger. Contour plots are not ideal to visualize small populations, so I would encourage pseudocolor plots with larger dot size. The gates are sometimes not consistently drawn for each sample (eg Fig. 1A OX-40), which must be done.
In Fig. 1 and Fig. 3 only marker-single-positive quantifications are performed. Often, only through combination of markers a specific staining can be reached (see many other SARS-CoV-2 T cell papers). Quantifications in Fig. 1, 2 and 3 are not done for each marker comparing the different stimulation conditions side-by-side (the focus is on comparing ECV and LCV). The authors should insert a comparison of the stimulation conditions for each marker and ECV vs. LCV in the supplementary data, so the stimulation effect can be better assessed compared to controls. Can the authors explain why there is so little activation after P/I treatment? Statistically non-significant differences should be labelled as such or, alternatively, it should be stated in the figure legend that any comparison that is not labelled with a p value is statistically non-significant. In the figure legend for each data set the exact number of individuals should be stated (n = x individuals) and it should be stated what the boxes and error bars around the geometric mean indicate.
Fig. 2: The authors should also include the data for the quadruple cytokine positive cells.
Fig. 4: The antibody data used for this figure are not explained in the methods section, nor are they shown as results. Both should be done. The correlations seem overall pretty homogenously distributed. Cluster analysis should be performed to identify co-correlations.
In the discussion the role of different expression levels of markers are discussed (eg “elevation” of CD38 or CD69). In the manuscript, cells were only classified as marker-positive or -negative. No quantitative expression levels were defined. So for each quantification, the MFIs of markers (pregated on the marker-positive population) should be quantified in terms of geometric mean of expression levels per individual.
The discussion should encompass a section describing the limitations of the study. One major limitation is that individual patients have not been followed up longitudinally for the two time points, which would have made the time point-comparisons more meaningful. Another limitation is that based on stimulation and marker expression, it cannot be differentiated between the mere presence of antigen-specific T cells and the quantity of functional T cells. In other words, at early time point after convalescence, the same number of antigen-specific T cells may be present, only that they are not reacting to the stimulation as much and are therefore partly being missed in this functional readout. HLA class II tetramers are a way to define the mere presence of antigen reactive T cells. In any case, the surprising “enhancement” of functional cells at late time points needs to be discussed against the background that you would expect contraction of antigen-specific T cells over time.
Language: The entire manuscripts needs careful revision with regards to language, there are several instances when eg sentences are incomplete.
Author Response
Reviewer 2
This is manuscript investigates the activation marker, cytokine and effector mediator profile of SARS-CoV-2 reactive T cells early and late after infection. A major aim of the study was to investigate the profile of reactive T cells with different markers in order to enable a systematic overview of the different functionalities. The major observation is that late after infection, there are more functional T cells compared to early after infection. In principle, this observation is surprising, given the fact that you would expect and many other groups have shown that there is peak T cell expansion within the first month after infection, and contraction thereafter. In this regard, it is noteworthy that it is important to differentiate between the mere presence of antigen specific T cells and the detection of T cells that are able to react to antigen stimulation (the latter may comprise a subpopulation of the former, which could explain the findings). This is particularly important for CD4 T cells, for which MHC multimers are less widely used compared to CD8 T cells, which can mark antigen specific T cells irrespective of functionality. An additional important limitation of the study includes the fact that it is not the same individuals who are followed up longitudinally. While the data are in principle of interest to the scientific community, the manuscript in its present needs to be revised considerably before it is suitable for publication.
Specific points:
1.The entire story would be more convincing if the same “enhancement” of functionality could be shown longitudinally for the same individuals. To interpret the “enhancement” of functionality, it would be important to know the actual time points from which the samples were taken. This should be reported for both the early and late time point.
The introduction is not focused in a way that it introduces specifically the research question that was experimentally tackled by the authors, which is a) wider polyfunctionality of CD4 T cells (more markers) b) short-term vs. long-term phenotype and c) investigation of T cell responses to SARS-2 in a non-high-income country. A first open question that is raised in the introduction is whether T cells are protective or can mediate immunopathology (lines 36-39), but this is not investigated in this study. Then it is mentioned that recent studies have only looked at limited markers (lines 45-48), but out of a gigantic pool of T cell studies on SARS-2 only two studies (ref 8, 10) are mentioned. SARS-2 reactive T cells have been in-depth characterized by scRNA seq (see eg Meckiff et al. Cell 2020, Bacher et al. Immunity 2020 or Fischer et al. Nat Comm 2021) and for a plethora of cytokines eg in Rydyznski Moderbacher et al. Cell 2020, to name a few. See Sette & Crotty et al. Cell 2021 for a review of the T cell response to SARS-2. The authors therefore should tone down their statement that they “looked in to a wider panel of T cell mediated immune responses” (line 48). The authors should also correct the false statement that “very limited studies have characterized the SARS-CoV-2-specific T cell response in COVID-19” (lines 52-53).
Reply: We thank reviewer for the valuable comment as suggested we have now added information related to our objective and also cited the reviewer suggested reference and modified the sentence accordingly. Lines 62-65, 67-70
2.Fig. 1A, 2A, 3A. These plots are still hard to read. Percentages should be larger. Contour plots are not ideal to visualize small populations, so I would encourage pseudocolor plots with larger dot size. The gates are sometimes not consistently drawn for each sample (eg Fig. 1A OX-40), which must be done.
Reply: As suggested we have now changed the representation plots in pseudocolor. Figures 1-3
3.In Fig. 1 and Fig. 3 only marker-single-positive quantifications are performed. Often, only through combination of markers a specific staining can be reached (see many other SARS-CoV-2 T cell papers).
Reply : In our cohort for the surface markers we see better expression only with single producing cell so we have retained the same for our analysis
4.Quantifications in Fig. 1, 2 and 3 are not done for each marker comparing the different stimulation conditions side-by-side (the focus is on comparing ECV and LCV). The authors should insert a comparison of the stimulation conditions for each marker and ECV vs. LCV in the supplementary data, so the stimulation effect can be better assessed compared to controls.
Reply: As suggested we have done separate statistical analysis to see the stimulation effect for each study group and individual markers this information is now added in the supplemental table. Sup. Table 1-3 and Lines 607-611
5.Can the authors explain why there is so little activation after P/I treatment?
Reply: We apologize for the lack of clarity, we have now added the antigen stimulation effect as supplemental table in that we notice there is a fivefold increase in the immune markers upon PMA/I situation compared to baseline, where as only 2-3 fold increase was seen upon SARS-CoV-2 antigens in comparison to baseline. Sup. Table 1-3 and Lines 607-611
6.Statistically non-significant differences should be labelled as such or, alternatively, it should be stated in the figure legend that any comparison that is not labelled with a p value is statistically non-significant. In the figure legend for each data set the exact number of individuals should be stated (n = x individuals) and it should be stated what the boxes and error bars around the geometric mean indicate.
Reply : We have now added this details in the figure legends. Line 565, 581, 596
7.Fig. 2: The authors should also include the data for the quadruple cytokine positive cells.
Reply: We thank reviewer for the comments for the quadruple cytokine most values are at very low frequency i.e., 0.001 were we unable do any statistical comparison between the groups
8.Fig. 4: The antibody data used for this figure are not explained in the methods section, nor are they shown as results. Both should be done.
Reply : We appreciate this comment and have now included details in the methods section. Lines 138 -150
9.The correlations seem overall pretty homogenously distributed. Cluster analysis should be performed to identify co-correlations.
Reply : As suggested we have now added the cluster analysis along with the existing correlation figure. Figure 4 and Lines 226-228
10.In the discussion the role of different expression levels of markers are discussed (eg “elevation” of CD38 or CD69). In the manuscript, cells were only classified as marker-positive or -negative. No quantitative expression levels were defined. So for each quantification, the MFIs of markers (pregated on the marker-positive population) should be quantified in terms of geometric mean of expression levels per individual.
Reply: We apologize for the lack of clarity, in our manuscript the all the reported data are quantitative expression of percentage (Frequency) level per individual. Line 135-136
11.The discussion should encompass a section describing the limitations of the study. One major limitation is that individual patients have not been followed up longitudinally for the two time points, which would have made the time point-comparisons more meaningful. Another limitation is that based on stimulation and marker expression, it cannot be differentiated between the mere presence of antigen-specific T cells and the quantity of functional T cells. In other words, at early time point after convalescence, the same number of antigen-specific T cells may be present, only that they are not reacting to the stimulation as much and are therefore partly being missed in this functional readout. HLA class II tetramers are a way to define the mere presence of antigen reactive T cells. In any case, the surprising “enhancement” of functional cells at late time points needs to be discussed against the background that you would expect contraction of antigen-specific T cells over time.
Reply: As suggested we have added this information in the discussion. Lines 312-314
12.Language: The entire manuscripts needs careful revision with regards to language, there are several instances when eg sentences are incomplete.
Reply : We apologize for the errors we have now modified it accordingly. Multiple Lines

Reviewer 3 Report
In this paper, the authors characterized the SARS-CoV-2 specific CD4+ T cell responses in late convalescent individuals and early convalescent individuals with or without SARS-CoV-2 specific antigens stimulation. And identify the correlation between SARS-CoV-2 specific antibodies and other innate cells in individuals with COVID-19. The result show frequencies of SARS-CoV-2-specific T cells of late convalescent individuals were higher than those of early convalescent individuals. And there maybe have a certain relationship between frequencies of CD4+ T cell and SARS-CoV-2 specific antibodies.
The manuscript is a topic of interest to the researchers in the related areas.The overall structure of the article is complete and the logic is clearer. However, there are also some descriptive accuracies as well as formatting problems, which needs to be revised.
Please check the attachment.

Author Response
Reviewer 3
In this paper, the authors characterized the SARS-CoV-2 specific CD4+ T cell responses in late convalescent individuals and early convalescent individuals with or without SARS-CoV-2 specific antigens stimulation. And identify the correlation between SARS-CoV-2 specific antibodies and other innate cells in individuals with COVID-19. The result show frequencies of SARS-CoV-2-specific T cells of late convalescent individuals were higher than those of early convalescent individuals. And there maybe have a certain relationship between frequencies of CD4+ T cell and SARS-CoV-2 specific antibodies.
The manuscript is a topic of interest to the researchers in the related areas.The overall structure of the article is complete and the logic is clearer. However, there are also some descriptive accuracies as well as formatting problems, which needs to be revised. The specific recommendations are as follows:
1.In the Discussion section, only CD69, CD38 and OX40 were discussed. The other cytokines mentioned in the paper, such as IFNγ, IL-2, TNFα, IL-17A, and so on, should be discussed, too.
Reply: We appreciate this comment and have now included details in the discussion section. Lines 268-278
2.Line 48: “looked in to” should be “looked into”.
Reply : As suggested we have now modified accordingly
3.Line 66: Add days after 180.
Reply : As suggested we have now modified accordingly. Multiple lines
4.Line 80: “ 2 × 106 cells/ml” should be “2 × 106 cells/mL”. “L” should be capitalized when indicating the unit of measurement. Please check the full manuscript and confirm.
Reply : As suggested we have now modified accordingly. Line. 102
5.Line 87 and Line 88: These sentences are incomplete. “SARS-CoV-2 whole cell lysate was (from BEI resources, NIAID, NIH)” and “ SARS-CoV-2 whole cell lysate 10 μg/mL”\
Reply : As suggested we have now modified accordingly. Lines 110-111
6.Line 123: Which methods were used to evaluate IgG, neutralizing antibodies and IgM, and how to determine positive and negative.
Reply : As suggested we have now added this information in the manuscript. Lines 138-150
7.Line 265: There are two “we”. Delete one of them.
Reply : As suggested we have now modified accordingly
8.Table 1: The text in the table should be aligned up and down. And last line, “(6.6%0” should be “ (6.6%)”.
Reply : As suggested we have now modified accordingly. Table 1
9.Table 2: The thickness of lines in the table is different. Please check.
Reply : As suggested by Reviewer 1 Table 2 was removed from the manuscript
- Fig.1B-1F: They are better to make into a combination diagram according to Fig.1A, and keep the same size and scale. The example is as follows.
Reply : As suggested we have now modified the figures accordingly.

Round 2
Reviewer 1 Report
Accept in present form
Author Response
We would like to thank the reviewer for insightful comments and suggestionsReviewer 2 Report
The manuscript has improved through the revision process. A few points remain:
- I disagree with reviewer 1 with regards to how quantified data should be displayed. I would strongly recommend a visualization that shows individual data points, like the authors did before.
- The authors should present the data from suppl. tables 1-3 as figures so the distribution of the data can be better assessed.
- The described limitations (discussion) should include the fact that the authors cannot distinguish between the presence and quality of antigen-specific cells (i.e. at the acute phase there could be more antigen-specific cells present than at late stages, but the response to stimulation could be different eg due to cellular phenotype and then the fraction of cells expressing cytokines would be influenced by that).
- In the abstract and in the last sentence of the introduction the authors should mention that they analysed data from a low-to-middle-income country, thereby highlighting a central strength of the study.
Author Response
The manuscript has improved through the revision process. A few points remain:
1. I disagree with reviewer 1 with regards to how quantified data should be displayed. I would strongly recommend a visualization that shows individual data points, like the authors did before.
Reply: As suggested we will retain the original submitted figures showing the scatter plots. Shown in Figures 1-3.
2. The authors should present the data from suppl. tables 1-3 as figures so the distribution of the data can be better assessed.
Reply : As suggested we have presented this data figures format. Supplemental Figures 1-3.
3. The described limitations (discussion) should include the fact that the authors cannot distinguish between the presence and quality of antigen-specific cells (i.e. at the acute phase there could be more antigen-specific cells present than at late stages, but the response to stimulation could be different eg due to cellular phenotype and then the fraction of cells expressing cytokines would be influenced by that).
Reply: We thank reviewer for the comment, as suggested we have now modified in the discussion.
3. In the abstract and in the last sentence of the introduction the authors should mention that they analysed data from a low-to-middle-income country, thereby highlighting a central strength of the study.
Reply: As suggested we have now modified accordingly.
